# Input Normalized Stochastic Gradient Descent Training for Deep Neural Networks

**Salih Atici, Hongyi Pan, Ahmet Enis Cetin**
**Department of Electrical and Computer Engineering**
**University of Illinois Chicago**
**Chicago, Illinois, USA**
`{hpan21, satici2, aecyy}@uic.edu`

Reviewed on OpenReview: `https://openreview.net/forum?id=5TaBxctwRZ&`

## Abstract

In this paper, we propose a novel optimization algorithm for training machine learning models called Input Normalized Stochastic Gradient Descent (INSGD), inspired by the Normalized Least Mean Squares (NLMS) algorithm used in adaptive filtering. When training complex models on large datasets, choosing optimizer parameters, particularly the learning rate, is crucial to avoid divergence. Our algorithm updates the network weights using stochastic gradient descent with $\ell_1$ and $\ell_2$-based normalizations applied to the learning rate, similar to NLMS. However, unlike existing normalization methods, we exclude the error term from the normalization process and instead normalize the update term using the input vector to the neuron. Our experiments demonstrate that our optimization algorithm achieves higher accuracy levels compared to different initialization settings. We evaluate the efficiency of our training algorithm on benchmark datasets using a toy neural network and several mature modern deep networks including ResNet-20, ResNet-50, MobileNetV3, WResNet-18, and Vision Transformer. Our INSGD algorithm improves ResNet-20's CIFAR-10 test accuracy from 92.57% to 92.67%, MobileNetV3's CIFAR-10 test accuracy from 90.83% to 91.13%, WResNet-18 on CIFAR-100 from 78.24% to 78.47%, and ResNet-50's accuracy on ImageNet-1K validation dataset from 75.60% to 75.92%.

## 1 Introduction

Deep Neural Networks (DNNs) have gained immense popularity and have been extensively applied across various research fields due to their convenience and ease of use in many machine learning tasks LeCun et al. (1995); He et al. (2016); Krizhevsky et al. (2017); Simonyan & Zisserman (2014); Long et al. (2015). Researchers from different domains can readily utilize DNN models for their work, as these models can adapt their parameters to find the best possible solutions for a wide range of problems, particularly in supervised learning scenarios. The parameters of a DNN model are updated using various optimization algorithms, and researchers have proposed different algorithms that offer fresh perspectives and address different conditions Ruder (2016). It is important to note that different optimization algorithms can yield different results in a given problem depending on the task at hand.

Stochastic Gradient Descent (SGD) is a widely adopted optimization algorithm for supervised learning in DNN models. It is a simple, efficient, and parallelizable algorithm that can produce very accurate results on large-scale datasets appropriate initial conditions Bottou (2010). Another popular algorithm Adam can outperform the SGD in some cases Kingma & Ba (2014). However, its initial learning rate is crucial. A relatively high value can lead to divergence. Optimization algorithms always play a significant role in training DNN models, but ensuring convergence of weights and finding the optimal solution for a given problem is not always guaranteed. Therefore, the evaluation of an optimization algorithm should also consider its limitations. Robustness to variability is a crucial attribute expected from any optimization algorithm.

Given the inherent ability of the Normalized Least Mean Squares (NLMS) algorithm to exhibit resilience to environmental fluctuations, it is plausible to anticipate that an optimization algorithm derived from NLMS principles would demonstrate favorable performance characteristics in dynamic settings or uncertain conditions. In this paper, we propose a novel optimization algorithm called Input Normalized Stochastic Gradient Descent (INSGD), which draws inspiration from the Normalized Least Mean Squares (NLMS) algorithm used in adaptive filtering Mathews & Xie (1993); Chan & Zhou (2010). Our study focuses on demonstrating the innovation from NLMS to INSGD and the effectiveness of INSGD over various tasks.

The organization of the paper is as follows. Sections 1.1 and 1.2 review the SGD and NLMS, respectively. Section 2 introduces the proposed Input Normalized Stochastic Gradient Descent (INSGD) algorithm. Section 3 presents simulation examples. Section 4 draws our conclusion.

## 1.1 Stochastic Gradient Descent

Stochastic Gradient Descent (SGD) is an iterative optimization method commonly used in machine learning to update the weights of a neural network model. It calculates the gradient of the weights based on the objective function defined to measure the error in the training phase and estimates the new set of weights using the gradients with a predefined step size. The SGD with a convex loss function can converge to the optimal or sub-optimal set of weights with the correct initial settings Li & Orabona (2019). The gradual convergence provided by gradient descent boosts optimizing the weights for any type of machine learning model.

Assume a pair of $(\mathbf{x}, \mathbf{y})$ composed of an arbitrary input $\mathbf{x}$ and an output $\mathbf{y}$. Given a set of weights $\mathbf{w} \in \mathbb{W}$ where $\mathbb{W}$ stands for the space of possible weights, a machine learning model predicts the output using a non-linear function $f(\mathbf{x}, \mathbf{w})$ and the optimal weights, $\mathbf{w}^*$, to minimize the objective (loss) function $L(\mathbf{y}, f(\mathbf{x}, \mathbf{w}))$:

$$\mathbf{w}^* = \arg \min_{\mathbf{w} \in \mathbb{W}} L(\mathbf{y}, f(\mathbf{x}, \mathbf{w})). \tag{1}$$

Due to the highly complex and non-linear nature of machine learning models, it is impossible to find a closed-form solution for the optimization problem given in Eq. (1) ada (2008). The stochastic gradient descent algorithm is introduced to avoid extensive computation and give an iterative method to estimate the optimal weights. The formula for SGD is given as:

$$\mathbf{w}(k+1) = \mathbf{w}(k) - \lambda \nabla_{\mathbf{w}(k)} L(\mathbf{y}_i, f(\mathbf{x}_i, \mathbf{w})), \tag{2}$$

where $\mathbf{w}(j)$ represents the weights at $j^{th}$ step, $\nabla_{\mathbf{w}(k)} L$ is the gradient of the objective function calculated using a single training example $(\mathbf{x}_i, \mathbf{y}_i)$, and $\lambda$ is determined by the step size and the learning rate.

Although SGD is a simple algorithm that can be applied to various tasks, it faces challenges related to tuning and scalability, which hinder its ability to converge quickly in deep learning algorithms. If the initial weights are not properly defined, or without preconditioned gradients that consider curvature information, the algorithm can get trapped in a local minima Le et al. (2011); Hinton & Salakhutdinov (2006). To estimate the minimum of the objective function more effectively, a deeper understanding of the error surface is required. In addition to using gradients, the exploitation of second-order derivatives can lead to faster convergence. However, this requires calculating the Hessian matrix of the objective function. Calculating the second derivative with respect to each weight is computationally expensive and can lead to memory issues in deep networks. The Hessian matrix and its approximations are also utilized in the Normalized Least Mean Squares (NLMS)-type methods, which will be discussed in the following subsection.

## 1.2 Normalized Least Mean Squares (NLMS)

As pointed out above, the NLMS is widely used to estimate the weights of an adaptive filter, which is a basic linear neuron. In minimum mean square error filtering, assume $\mathbf{u}$, the input to a system, is a $1 \times M$ random vector with zero mean and a positive-definite covariance matrix $\mathbf{R_u}$ and $d$, the desired output of the system, is a scalar random variable with zero mean and a finite variance $\sigma_d^2$. The linear estimation problem is defined as the solution of

$$\min_{\mathbf{w}} \mathbb{E} \left| d - \mathbf{u}\mathbf{w} \right|^2, \tag{3}$$

where $\mathbf{w}$ is a vector containing the filter coefficients to be optimized. The linear estimation problem declares the cost function as the mean-square error and it is defined as:

$$J(w) = \mathbb{E}\,|d - \mathbf{uw}|^2 = \mathbb{E}(d - \mathbf{uw})(d - \mathbf{uw})^T, \tag{4}$$

where $(.)^T$ denotes a transpose. If we expand Eq. (4), it is straightforward to obtain the cost function $J(w)$ in terms of the covariance and cross-covariance matrices:

$$J(w) = \sigma_d^2 - \mathbf{R}_{d\mathbf{u}}^T\mathbf{w} - \mathbf{w}^T\mathbf{R}_{d\mathbf{u}} + \mathbf{w}^T\mathbf{R}_{d\mathbf{u}}\mathbf{w}, \tag{5}$$

where $\mathbf{R}_{d\mathbf{u}} = \mathbb{E}[d\mathbf{u}]$ is the cross-covariance matrix of $d$ and $\mathbf{u}$. The closed-form solution to such a problem in (3) can be found using the linear estimation theory as $\mathbf{R}_{\mathbf{u}}\mathbf{w}^o = \mathbf{R}_{d\mathbf{u}}$; however, it may not be possible to obtain a closed-form solution for problems with criteria other than the mean-square-error criterion.

The Least Mean Squares (LMS) algorithm Widrow et al. (1960) computes the stochastic gradient and updates the weight vector iteratively to find a solution for the problem in Eq. (3). The weight vector can be updated using the following iterative process:

$$\mathbf{w}(j) = \mathbf{w}(j-1) + \lambda\mathbf{u}^T(j)\mathbf{e}(j), \tag{6}$$

where $\mathbf{u}(j)$ is $j$-th observation of the random vector $\mathbf{u}$ and $\mathbf{e}(j) = \mathbf{d}(j) - \mathbf{u}^T\mathbf{w}(j-1)$ is the error vector at time $j$. The updating term is obtained as the negative of the stochastic gradient of the mean squared error function defined in Eq. (4) with respect to the weights.

The NLMS algorithm has been shown to achieve a better convergence rate compared to LMS by incorporating a different step-size parameter for each component $\mathbf{u}_i$ of the vector $\mathbf{u}$ Sayed (2008). The LMS algorithm can encounter scalability issues when the input signal is large or when the step-size parameter is too large. Since the LMS algorithm uses a gradient-based approach to update the filter coefficients, and if the step-size parameter is too large, the filter coefficients can diverge. To address this, normalization is introduced to the update term:

$$\mathbf{w}(j) = \mathbf{w}(j-1) + \lambda\frac{\mathbf{e}(j)}{||\mathbf{u}(j)||_2^2}\mathbf{u}(j), \tag{7}$$

and the NLMS converges to the Wiener filter solution of the optimization problem in (3) as long as $0 < \lambda < 2$ Theodoridis et al. (2010); Yamada et al. (2002).

Another interpretation of the NLMS algorithm is based on the fact that the error $\mathbf{e}(j) = \mathbf{d}(j) - \mathbf{u}^T\mathbf{w}$ should be minimized by selecting an appropriate weight vector $\mathbf{w}$. The equation $\mathbf{d}(j) = \mathbf{u}^T\mathbf{w}$ is a hyperplane in the $M$ dimensional weight space $\mathbf{w} \in \mathbb{R}^M$. When the vector $\mathbf{w}(j-1)$ is projected onto the hyperplane $\mathbf{e}(j) = \mathbf{d}(j) - \mathbf{u}^T\mathbf{w}$, we obtain the update equation:

$$\mathbf{w}(j) = \mathbf{w}(j-1) + \frac{\mathbf{e}(j)}{||\mathbf{u}(j)||_2^2}\mathbf{u}(j). \tag{8}$$

As shown in Fig. 1, the error is minimized by selecting the next weight vector on the hyperplane $\mathbf{d}(j) = \mathbf{u}^T\mathbf{w}$. The orthogonal projection operation described in Eq. (8) minimizes the Euclidean distance between the vector $\mathbf{w}(j-1)$ and the hyperplane $\mathbf{d}(j) = \mathbf{u}^T\mathbf{w}$ Combettes (1993); Trussell & Civanlar (1984); Cetin et al. (1997; 2013) The weights converge to the intersection of the hyperplanes as shown in Fig. 1, provided that the intersection of the hyperplanes is non-empty Combettes (1993); Cetin et al. (2013).

Other distance measures lead to different update equations such as the $\ell_1$-norm-based updates:

$$\mathbf{w}(j) = \mathbf{w}(j-1) + \frac{\mathbf{e}(j)}{||\mathbf{u}(j)||_1}\mathbf{u}(j), \tag{9}$$

where $||\mathbf{u}(j)||_1$ is the $\ell_1$ norm of the vector $\mathbf{u}_j$ Gunay et al. (2012); Sayin et al. (2014); Arikan et al. (1994; 1995); Aydin et al. (1999). The $\ell_1$-norm-based method is usually more robust to outliers in input.

This paper describes a new optimization algorithm inspired by Normalized LMS. It is called Input Normalized-SGD (INSGD) and utilizes the same approach as in NLMS. INSGD provides a better solution to the variability issue that may cause divergence or inconsistent results and obtains better accuracy results on benchmark datasets. By adapting the concepts of NLMS to deep learning, we can potentially improve the convergence behavior and overall performance of DNN models.

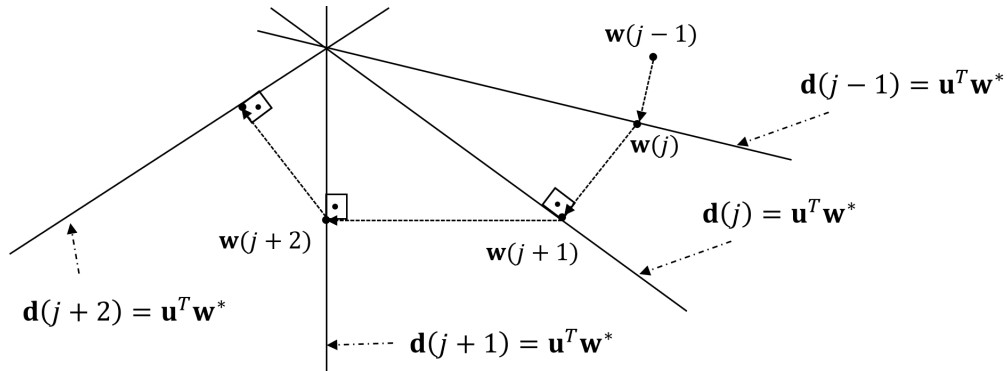

Figure 1: Geometric description of the NLMS projection in $\mathbb{R}^2$.

## 2 Related Works

Machine learning models commonly utilize the backpropagation method for optimization Rumelhart et al. (1986); LeCun et al. (1988). Stochastic Gradient Descent (SGD) is a widely used optimization algorithm with various modifications in the machine learning community. While SGD can provide convergence with proper initialization, researchers have identified both positive and negative aspects of SGD and have attempted to enhance it according to their specific objectives Ruder (2016). One issue with SGD is that it updates weights based solely on the instantaneous gradient, which may lead to a lack of global information and oscillations. Another challenge is using a constant learning rate for all weights in the model. As training progresses, certain weights become more important than others, requiring different step sizes to ensure effective learning.

In recent years, the Adaptive Gradient (AdaGrad) algorithm aims to enhance optimization by adaptively adjusting the learning rate for each weight based on the cumulative sum of past and current squared gradients Duchi et al. (2011). This adaptive approach allows for finer adjustments of the learning rate, ensuring larger updates for weights with smaller gradients and vice versa. AdaGrad's formulas are:

$$\mathbf{w}(k+1) \leftarrow \mathbf{w}(k) - \frac{\gamma}{\sqrt{v(k)+\epsilon}} \nabla_{\mathbf{w}(k)} L, \tag{10}$$

$$v(k) \leftarrow v(k-1) + \left[\nabla_{\mathbf{w}(k)} L\right]^2, \tag{11}$$

where $v$ represents the weighted moving average of squared gradients, $\gamma$ is the learning rate, and $v(-1) = 0$.

Later, RMSProp builds upon AdaGrad by incorporating momentum through an exponentially weighted moving average of squared gradients Hinton et al. (2012). This modification addresses AdaGrad's issue of diminishing learning rates, ensuring a smoother and more stable update process. RMSProp's formulas are:

$$\mathbf{w}(k+1) \leftarrow \mathbf{w}(k) - \frac{\gamma}{\sqrt{v(k)+\epsilon}} \nabla_{\mathbf{w}(k)} L, \tag{12}$$

$$v(k) \leftarrow \beta v(k-1) + (1-\beta) \left[\nabla_{\mathbf{w}(k)} L\right]^2, \tag{13}$$

where $\beta$ represents the momentum parameter. RMSProp strikes a balance between AdaGrad's adaptability and momentum's stability, leading to improved optimization performance.

Another widely used optimization algorithm is Adaptive Moment Estimation (Adam) Kingma & Ba (2014). It builds upon the concepts of momentum and the divisor factor used in RMSProp. In addition to maintaining an exponentially weighted moving average of the squared gradients like RMSProp, Adam also incorporates the notion of momentum by keeping track of an exponentially weighted moving average of the gradients themselves. This combination of momentum and the divisor factor makes Adam more adaptive and robust

compared to RMSProp and AdaGrad. By considering both the first and second moments of the gradients, Adam adjusts the learning rate for each parameter individually, taking into account both the magnitude and direction of the gradients. This enables Adam to converge faster and handle a wider range of optimization scenarios. The algorithm is implemented as:

$$\mathbf{w}(k+1) \leftarrow \mathbf{w}(k) - \frac{\gamma}{\sqrt{v(k) + \epsilon}} \mathbf{m}(k), \tag{14}$$

$$v(k) \leftarrow \beta v(k-1) + (1 - \beta) \left[ \nabla_{\mathbf{w}(k)} L \right]^2, \tag{15}$$

$$\mathbf{m}(k) \leftarrow \beta \mathbf{m}(k-1) + (1 - \beta) \nabla_{\mathbf{w}(k)} L, \tag{16}$$

where $\mathbf{m}$ is the momentum and $\mathbf{m}(-1) = \mathbf{0}$.

Layer-wise Adaptive Rate Scaling (LARS) is an optimization algorithm designed to improve the training of deep neural networks by dynamically adjusting the learning rates for each layer based on the norm of the gradients and the norm of the weights You et al. (2017):

$$\mathbf{w}(k+1) \leftarrow \mathbf{w}(k) - \eta \cdot \frac{\|\mathbf{w}(k)\|}{\|\nabla_{\mathbf{w}(k)} L\| + \epsilon} \cdot \nabla_{\mathbf{w}(k)} L \tag{17}$$

where $\theta_t$ is the parameter vector at time step $t$, $\eta$ is the base learning rate, $\nabla L(\theta_t)$ is the gradient of the loss function with respect to $\theta_t$, $\epsilon$ is a small constant to prevent division by zero. The limitation is the performance of LARS compared to other adaptive methods can vary and is influenced by factors such as the dataset, model architecture, and hyperparameters.

Another adaptive learning algorithm proposed by Singh et al. (2015) presented a Layer-Specific Adaptive Learning Rate (LSALR), where the parameters in the same layer share similar gradients. Therefore, the learning rate of the entire layer should be similar but different layers should have different learning rates. The work is described to adjust the learning rate to escape from the saddle points and it uses the $\ell_2$ norm in gradients:

$$\mathbf{w}(k+1) \leftarrow \mathbf{w}(k) - \gamma \left( 1 + \log \left( 1 + \frac{1}{||\nabla_{\mathbf{w}(k)} L||_2} \right) \right) \nabla_{\mathbf{w}(k)} L. \tag{18}$$

Eq. (18) allows the learning rate increases when the gradients are small. The aim is to correct the update term when the gradients are small in the high error low curvature saddle points. Therefore, the algorithm escapes from saddle points with a large learning rate. Similarly, it scales the learning rate to stability if the gradients are too large. The use of the *log* function provides the scaling under different conditions.

In summary, Adam, AdaGrad, and RMSProp are optimization algorithms that address the limitations of standard stochastic gradient descent (SGD). These algorithms improve the convergence speed in various scenarios. While they incorporate normalization parameters, the update terms in these algorithms are still input-dependent and gradually decrease over iterations. On the other hand, our approach uses a normalization term based on the layer's input. its details will be presented in the following section.

## 3 Methodology

### 3.1 Input Normalized Stochastic Gradient Descent Algorithm

Input Normalized Stochastic Gradient Descent (INSGD) utilizes a similar approach as NLMS. We focus on enhancing the robustness to variability and real-time processing capabilities of the INSGD optimizer, addressing challenges related to various training settings.

In deep learning, we minimize the cost function:

$$F(\mathbf{W}) = \frac{1}{N} \sum_{k=1}^{N} F_k(\mathbf{W}),$$

where $\mathbf{W}$ represents the parameters of the network, $N$ is the number of training samples, and $F_k(\mathbf{W})$ is the loss due to the $k$-th training data. Let us first assume that there are linear neurons in the last layers of the network and $d_i$ is the desired value of the $i$-th neuron. Furthermore, let $\mathbf{w_{i,0}}$ be the initial weights of the $i$-th neuron. We want the neuron to satisfy

$$d_i = \mathbf{w} \cdot \mathbf{x},$$

where $\mathbf{x}$ denotes the input vector to the neuron. During training, we have $\mathbf{w_{i,0}} \cdot x_k \neq d_i$ where $\mathbf{x}_k$ is the input vector due to the $k$-th training pattern. We select the new set of weights of the neuron by solving

$$\arg\min_{\mathbf{w}} ||\mathbf{w}_{i,0} - \mathbf{w}||^2, \tag{19}$$

$$\text{s.t. } \mathbf{w} \cdot \mathbf{x}_k = d_i.$$

One can easily obtain the solution using the Lagrange multiplier method, and the solution to the optimization problem is the orthogonal projection onto the hyperplane $\mathbf{w} \cdot \mathbf{x}_k = d_i$. Solving Eq.(19) gives us an update equation

$$\mathbf{w}_{i,1} = \mathbf{w}_{i,0} + \lambda \frac{e_i}{\epsilon + ||\mathbf{x}_k||^2} \mathbf{x}_k, \tag{20}$$

where the error $e_i = d_i - \mathbf{w}_{i,0} \cdot \mathbf{x}_k$, the update parameter $\lambda = 1$, and $\epsilon$ is a small number to avoid the division by 0. This selection of weights reduces $F_k(\mathbf{W})$ and it is the same as the gradient descent with a new step size determined by the length of the input vector. It is also the well-known NLMS algorithm used in adaptive filtering and signal processing as shown in Sec 1.2, Eq. (7). The NLMS algorithm converges for $0 < \lambda < 2$ when the input is a wide-sense stationary random process. Inspired by the NLMS algorithm we can continue updating the neurons of the inner layers of the network in the same manner.

When the $i$-th neuron is not linear, we have

$$\psi(\mathbf{w} \cdot \mathbf{x}) = d_i, \tag{21}$$

where $\psi(\cdot)$ is the activation function. In this case, we solve the following problem to update the neuron weights:

$$\arg\min_{\mathbf{w}} ||\mathbf{w}_{i,0} - \mathbf{w}||^2, \tag{22}$$

$$\text{s.t. } \psi(\mathbf{w} \cdot \mathbf{x}_k) = d_i,$$

or

$$\arg\min_{\mathbf{w}} ||\mathbf{w}_{i,0} - \mathbf{w}||^2, \tag{23}$$

$$\text{s.t. } \mathbf{w} \cdot \mathbf{x}_k = \phi(d_i),$$

where $\phi(\cdot)$ is the inverse of the $\psi(\cdot)$ function. When $\psi(\cdot)$ is the sigmoid, leaky-RELU, or tanh, $\psi(\cdot)$ has a well-defined inverse. If the activation function is ReLU, the negative values in the inverse are set to 0s. In this case, the weight update equation will be

$$\mathbf{w}_{i,1} = \mathbf{w}_{i,0} + \lambda \frac{(\phi(d_i) - \mathbf{w}_{i,0} \cdot \mathbf{x}_k)}{\epsilon + ||\mathbf{x}_k||^2} \mathbf{x}_k. \tag{24}$$

By employing the solution described in Eq. (24), the NLMS algorithm can be adapted to optimize the weights in the final layer to minimize various cost functions. However, extending the INSGD algorithm to deeper networks with multiple layers poses a challenge in its derivation. We adopt similar assumptions to those used in the backpropagation algorithm to derive the INSGD algorithm for each weight in a deep-learning model. These assumptions provide a foundation for developing the INSGD algorithm, allowing effectively optimizing the weights across the layers of the deep learning model.

In addition to the final layer, we incorporate the input feature maps of each layer to apply the gradient term with normalization to the neurons using the backpropagation algorithm. This enables the optimizer to propagate the gradients and update the weights layer-wisely throughout the network. By leveraging the information from the input feature maps, we enhance the training process by ensuring that the gradients are

appropriately scaled and normalized at each layer. This approach allows for effective gradient propagation and weight updates, ultimately contributing to improved optimization and performance of the deep learning model:

$$\mathbf{w}_{k+1} = \mathbf{w}_k - \mu \frac{\nabla_{\mathbf{w}_k} L(\mathbf{e})}{\epsilon + ||\mathbf{x}_k||_2^2}, \tag{25}$$

where $\mathbf{x}_k$ is the vector of inputs to the neuron and $\mathbf{w}_k$ are the weights of the neurons. Note that we drop $i$ in the weight notation that represents the neuron since the algorithm is applicable to every neuron. For convenience, we also change the notation for the learning rate from $\lambda$ to $\mu$. A description of how the INSGD optimizer algorithm works for any layer of a typical deep network is shown in Fig. 2.

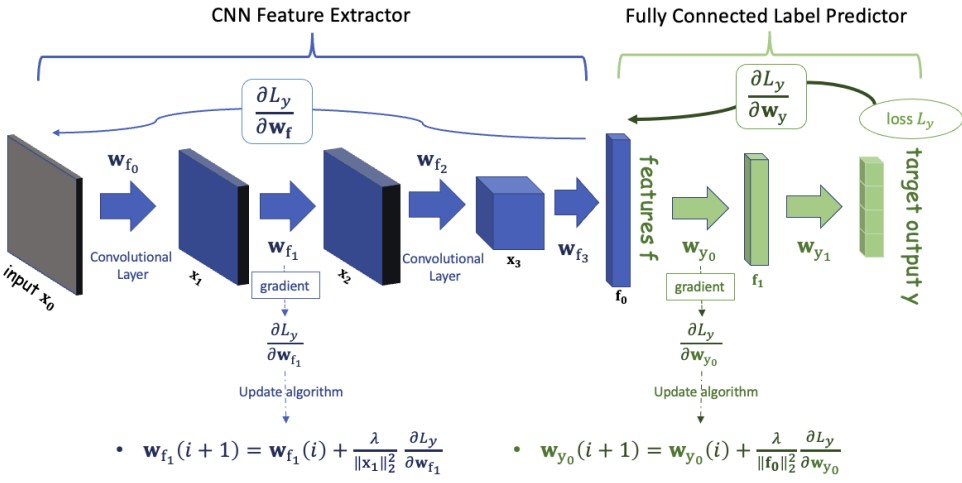

Figure 2: INSGD algorithm for different layers. It utilizes the input to each layer to update the weights.

Power normalization in NLMS is used to introduce a memory into the recursion so that the input power is estimated using the input data in the remote past. In the INSGD algorithm, we introduce an input momentum term to estimate the power of the dataset, enabling power normalization. By replacing the denominator term with the estimated input power, we emphasize the significance of power estimation in our algorithm. Furthermore, the utilization of input momentum allows capturing the norm of all the inputs. Denoted as $P$, the input momentum term accumulates the squared $\ell_2$ norm of the input instances:

$$P_k = \beta P_{k-1} + (1-\beta)||\mathbf{x}_k||_2^2. \tag{26}$$

While estimating the input power is crucial, the normalization factor can grow excessively, resulting in infinitesimally small updates. To address this, we draw inspiration from the Layer Specific Adaptive Learning Rate (LSALR) approach Singh et al. (2015) and employ the logarithm function and moving-average method to stabilize the normalization factor. However, the use of the logarithm function introduces the risk of negative values. If the power is too low, the function could yield a negative value, reversing the direction of the update. To mitigate this, we employ a function with the rectified linear unit, which avoids the issue of negative values. Adding a regularizer may not be sufficient to resolve this problem, hence the choice of the rectified linear unit function. The function is designed as follows:

$$f_\epsilon(u) = \begin{cases} u & \text{if } u \geq \epsilon, \\ \epsilon & \text{if } u < \epsilon, \end{cases} \tag{27}$$

where $\epsilon$ is a regularizer to avoid the division by 0. After devising the function in Eq. (27) and the logarithm approach for Eq. (26), the optimization algorithm for any weight in any layer in a network becomes

$$\mathbf{w}_{k+1} = \mathbf{w}_k - \frac{\mu}{f_\epsilon(\log(P_k))} \nabla_{\mathbf{w}_k} L(\mathbf{e}), \tag{28}$$

where $P_k$ is defined in Eq. (26), and it is the estimate of the input power that is updated with every instance of $\mathbf{x}$ and the proposed $\epsilon = 0.01$. Therefore, it makes sure that the update term is always positive and stable. The iterative algorithm defined in Eq. (28) is the Input Normalized SGD algorithm.

One can explore different norms, such as the $l_1$ or $l_\infty$ norm, as alternatives to the $l_2$ norm. In our experiments, we also investigated using the $l_1$ norm to assess its impact on performance. NLMS algorithms based on the $l_1$ norm are known to be more robust against outliers in the input, which suggests potential benefits in deep neural network training. In this study, we examined both the $l_2$ and $l_1$ norms and their implications. Since NLMS is based on the $l_2$ norm, the algorithm presented in Eq. (26) utilizes the $l_2$ norm. However, for broader applicability, we can adapt the power estimation as follows:

$$P_k = \beta P_{k-1} + (1 - \beta)||\mathbf{x}_k||_p^p \tag{29}$$

where $||.||_p^p$ is the $p$ power of the $p$-norm. Extension of the INSGD to convolutional layers is straightforward. The pseudocode algorithm of INSGD is given in Algorithm 1.

---

**Algorithm 1** Input Normalized Gradient Descent with Momentum

---

  **for** $t \leftarrow 1$ to ... **do**
    $g_t \leftarrow \nabla_\theta f_t(\theta_{t-1})$                                           ▷ Denote the gradient
    **if** $\beta \neq 0$ **then**                                ▷ If input momentum is not 0
        **if** $t > 1$ **then**
            $P_t \leftarrow \beta P_{t-1} + (1 - \beta)||\mathbf{x}_{t,\theta}||_2^2$              ▷ Accumulate the power of input norm
        **else**
            $P_t \leftarrow ||\mathbf{x}_{t,\theta}||_2^2$
        **end if**
    **end if**
    $g_t \leftarrow \dfrac{g_t}{f(\log(P_t))}$                           ▷ Division by input norm
    **if** $\lambda \neq 0$ **then**                                      ▷ Weight Decay
        $g_t \leftarrow g_t + \lambda\theta_{t-1}$
    **end if**
    **if** $\gamma \neq 0$ **then**                                 ▷ Gradient with Momentum
        **if** $t > 1$ **then**
            $\mathbf{b}_t \leftarrow \gamma\mathbf{b}_{t-1} + (1 - \tau)g_t$
        **else**
            $\mathbf{b}_t \leftarrow g_t$
        **end if**
        $g_t \leftarrow \mathbf{b}_t$
    **end if**
    $\theta_t \leftarrow \theta_{t-1} - \mu g_t$                                 ▷ Update the Weights
  **end for**

---

### 3.2 Models Architecture

In this study, we conduct experiments using six different networks to evaluate the performance of the INSGD algorithm in the classification tasks of CIFAR-10, CIFAR-100, and ImageNet-1K. We make modifications to the network architectures and initialization settings to assess the impact of the INSGD algorithm. In this study, we employ several networks for the classification tasks. Specifically, we utilize ResNet-20 He et al. (2016), MobileNetV3 Howard et al. (2019) and Vision Transformer (ViT) Dosovitskiy et al. (2020) for CIFAR-10, WResNet-18 Zagoruyko & Komodakis (2016) for CIFAR-100, ResNet-50 and MobileNetV3 for ImageNet-1K. Additionally, we design a custom CNN architecture specifically for CIFAR-10, which consists

of four convolutional layers, each followed by a batch normalization layer. These networks are chosen to provide a diverse set of architectures and enable a comprehensive evaluation of the INSGD algorithm's performance across different datasets. The structures of ResNet-20 and custom-designed CNN in this study are shown in Tables 1 and 2, respectively.In addition, we choose a patch size of 4, a linear layer dimension of 512, a depth of 6, and 8 heads in designing the Vision Transformer (ViT). We train it using warm restarts with Cosine annealing Loshchilov & Hutter (2016).

Table 1: ResNet-20 Structure for CIFAR-10 classification task. Building blocks are shown in brackets, with the numbers of blocks stacked.

| Layer | Output Shape | Implementation Details |
|---|---|---|
| Conv1 | $16 \times 32 \times 32$ | $3 \times 3, 16$ |
| Conv2_x | $16 \times 32 \times 32$ | $\begin{bmatrix} 3 \times 3, 16 \\ 3 \times 3, 16 \end{bmatrix} \times 3$ |
| Conv3_x | $32 \times 16 \times 16$ | $\begin{bmatrix} 3 \times 3, 32 \\ 3 \times 3, 32 \end{bmatrix} \times 3$ |
| Conv4_x | $64 \times 8 \times 8$ | $\begin{bmatrix} 3 \times 3, 32 \\ 3 \times 3, 64 \end{bmatrix} \times 3$ |
| GAP | 64 | Global Average Pooling |
| Output | 10 | Linear |

Table 2: Structure of the custom network with 4 conv layers for the CIFAR-10 classification task.

| Layer | Output Shape | Implementation Details |
|---|---|---|
| Conv1 | $8 \times 32 \times 32$ | $3 \times 3, 8$ |
| Conv2 | $16 \times 16 \times 16$ | $3 \times 3, 16, \text{stride} = 2$ |
| Conv3 | $32 \times 8 \times 8$ | $3 \times 3, 32, \text{stride} = 2$ |
| Conv4 | $64 \times 4 \times 4$ | $3 \times 3, 64, \text{stride} = 2$ |
| Dropout | $64 \times 4 \times 4$ | $p = 0.2$ |
| Flatten | 1024 | - |
| Output | 10 | Linear |

On large benchmark datasets, traditional optimization algorithms often struggle to find the optimum results if the learning rate is not properly chosen. In such cases, these algorithms may diverge and fail to converge to the desired solution. However, the INSGD algorithm offers a solution by providing flexibility in learning rate selection, thereby improving the chances of reaching the global optimum. By adapting the learning rate dynamically based on the input and gradient information, INSGD enhances the optimization process and increases the likelihood of achieving superior results on large-scale datasets.

## 4 Experimental Results

Our experiments are carried out on a workstation with an NVIDIA GeForce GTX 1660 Ti GPU for the CIFAR-10 and a workstation with an NVIDIA RTX A6000 GPU for the CIFAR-100 and ImageNet-1K.

### 4.1 CIFAR-10 Classification

We conduct a series of experiments using the CIFAR-10 dataset which consists of 10 classes, initially employing the custom-designed CNN and ResNet-20 models for training. In certain experiments, we make modifications to the custom network to explore the algorithm's capabilities. All the experiments are repeated 7 times with different seeds to verify the results. These experiments aim to assess the algorithm's performance under various conditions.

The base setting employed the SGD optimizer with a weight decay of 0.0005 and a momentum of 0.9. The models are trained using a mini-batch size of 128 for 200 epochs, with an initial learning rate ranging from 0.5 to 0.01. The learning rate is reduced at multiple steps with varying rates. To augment the data, we perform padding of 4 pixels on the training images, followed by random crops to obtain 32x32 images. Random horizontal flips are also applied to the images with a probability of 0.5. Normalization is performed on the images using a mean of [0.4914, 0.4822, 0.4465] and a standard deviation of [0.2023, 0.1994, 0.2010]. Throughout the training process, the best models are saved based on their accuracy on the CIFAR-10 test dataset. These settings are adopted from He et al. (2016).

In the initial experiment, we employ the ResNet-20 model as our baseline. The batch size is fixed at 128. We compare the accuracy results of our algorithm against those of other commonly used optimization algorithms, which are discussed in Section 2. The detailed accuracy results are presented in Table 3.

Table 3: Accuracy results of ResNet-20 on the CIFAR-10 dataset with different initial learning rates using different optimization algorithms.

| Optimizer | Initial Learning Rate | Test Accuracy |
|---|---|---|
| SGD | 0.1 | 92.57±0.11% |
| Adam | 0.001 | 91.34±0.01% |
| Adagrad | 0.1 | 89.41±0.01% |
| Adadelta | 0.1 | 89.24±0.01% |
| INSGD-$\ell_1$ | 0.1 | 92.67±0.13% |
| INSGD-$\ell_2$ | 0.1 | 92.60±0.11% |

In addition to showcasing the testing accuracy results, understanding the behavior of each optimizer throughout the training process is crucial. Figure 4 visually represents the progression of testing set errors for each optimizer over 200 epochs. By training the ResNet-20 model with each optimizer, we can observe the corresponding testing set error depicted in the plot. This visualization offers valuable insights into the convergence speed and overall behavior of each optimizer, enabling a comprehensive analysis of their performance and effectiveness.

As depicted in Figure 4, the INSGD algorithm with both norms consistently exhibits lower error rates in the testing set, which indicates the superior performance and effectiveness of INSGD in optimizing the model's parameters and minimizing the testing set errors. The ability of INSGD to adaptively adjust the learning rates for each individual parameter contributes to its remarkable performance in achieving lower errors during the training process. We can also observe that the behavior of INSGD closely aligns with that of SGD, indicating their compatibility and similarity in optimization behavior.

As seen in Table 3, only SGD closely matches the performance of our optimizer in this training scenario. Therefore, we concentrate on comparing INSGD with SGD for training ResNet-20 on CIFAR-10. We explore the impact of varying batch sizes on the normalization factor to understand how input size affects the training process. Analyzing the results across different batch sizes is crucial due to the trade-off between time and memory usage. While larger datasets may benefit from larger batch sizes to expedite training time, it is important to consider the increased memory requirements. Table 4 presents the accuracy results of other algorithms and INSGD when training the model with different batch sizes. To accommodate the increased batch size, we adjust the learning rate similar to the linear scaling rule described in Goyal et al. (2017).

Table 4 demonstrates that the INSGD optimizer maintains high performance across different batch sizes. Similar to NLMS, we anticipate that INSGD will be effective for real-time processing (online learning), highlighting its potential for superior performance in online learning scenarios.

To enhance the diversity of models utilized in our experiments, we incorporate the MobileNetV3 model for comparative analysis. A batch size of 256 is used in MobileNetV3 training. The mean and standard deviation of 7 experiments with different seeds are shown in Table 5. As depicted, the results clearly demonstrate that the INSGD algorithm outperforms other conventional optimization algorithms in terms of performance. This

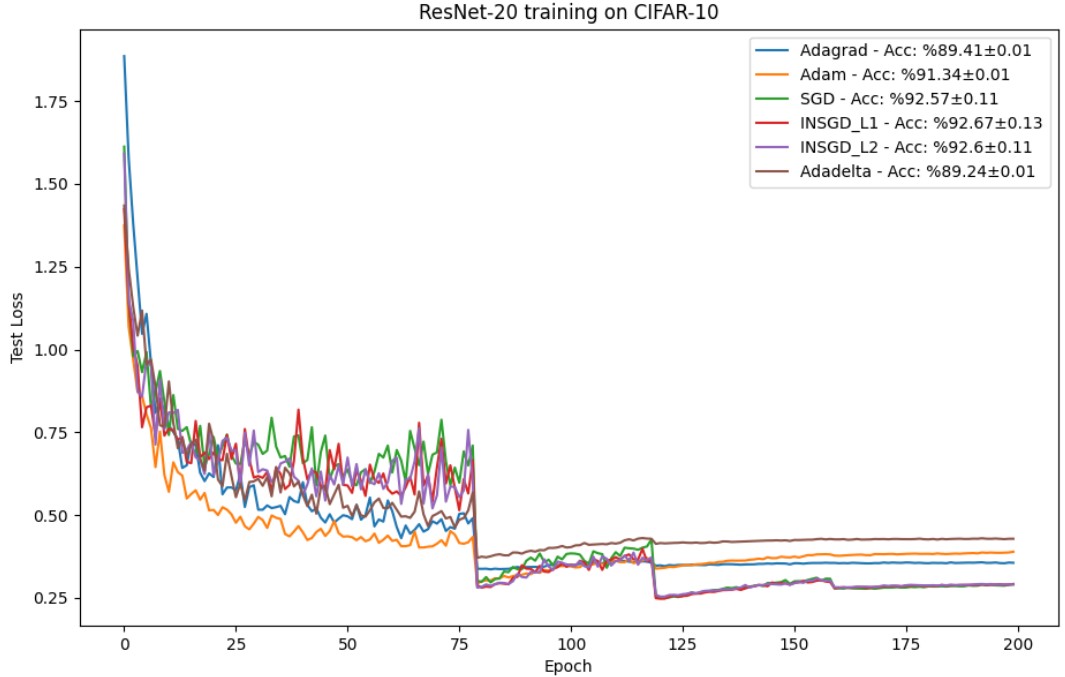

Figure 3: **Optimizer performance over 200 epochs**: Comparing test loss convergence across different optimizers.

Table 4: Accuracy results of the ResNet-20 on the CIFAR-10 dataset with different batch sizes.

| Optimizer | Batch | Learning Rate | Test Accuracy |
|:---:|:---:|:---:|:---:|
| SGD | 128 | 0.1 | 92.57±0.11% |
| INSGD-$\ell_1$ | 128 | 0.1 | 92.67±0.13% |
| INSGD-$\ell_2$ | 128 | 0.1 | 92.60±0.11% |
| SGD | 256 | 0.1 | 92.43±0.23% |
| INSGD-$\ell_1$ | 256 | 0.1 | 92.46±0.24% |
| INSGD-$\ell_2$ | 256 | 0.1 | 92.44±0.23% |
| SGD | 512 | 0.2 | 92.25±0.26% |
| INSGD-$\ell_1$ | 512 | 0.2 | 92.38±0.21% |
| INSGD-$\ell_2$ | 512 | 0.2 | 92.32±0.26% |

finding highlights the superior capabilities of INSGD in achieving improved outcomes across the evaluated metrics.

Table 5: Accuracy results of MobileNetV3 on the CIFAR-10 dataset with different initial learning rates using different optimization algorithms.

| Optimizer | Initial Learning Rate | Test Accuracy |
|---|---|---|
| Adam | 0.001 | 91.75±0.09% |
| Adagrad | 0.05 | 87.24±0.01% |
| Adadelta | 0.05 | 84.78±0.29% |
| SGD | 0.05 | 93.06±0.11% |
| INSGD-$\ell_1$ | 0.05 | 93.04±0.16% |
| INSGD-$\ell_2$ | 0.05 | 93.10±0.01% |

We also conduct experiments using the custom network for the CIFAR-10 training to validate our algorithm. We employed similar settings to those used in ResNet-20. The accuracy results of the custom network with different initial learning rates are presented in Table 6.

Table 6: Accuracy results of the custom-designed CNN on the CIFAR-10 dataset with different initial learning rates and reduction rates.

| Optimizer | Initial Learning Rate | Test Accuracy |
|---|---|---|
| SGD | 0.1 | 79.24±0.39% |
| INSGD-$\ell_1$ | 0.1 | 79.41±0.37% |
| INSGD-$\ell_2$ | 0.1 | 79.30±0.38% |
| SGD | 0.25 | 65.92±1.32% |
| INSGD-$\ell_1$ | 0.25 | 73.24±1.17% |
| INSGD-$\ell_2$ | 0.25 | 74.68±0.85% |
| SGD | 0.01 | 79.08±0.34% |
| INSGD-$\ell_1$ | 0.01 | 78.83±0.43% |
| INSGD-$\ell_2$ | 0.01 | 78.92±0.31% |

The toy network, used as a simplified representation of the model, plays a crucial role in evaluating the effectiveness of our algorithm. The results obtained from training the toy network confirm the robustness of INSGD to variability, as it shows consistent accuracy results regardless of the network architecture or the learning rate used. Given the overlap in the experiments conducted with the custom network and ResNet-20, we opted not to replicate the ResNet-20 experiments using the toy network. This decision was made to avoid redundancy in our findings and to focus on exploring the direct impact of INSGD.

We conducted experiments on a modified network architecture to explore the impact of INSGD's input normalization. Given that INSGD normalizes the input, we hypothesized that the batch normalization layer might become redundant. To test this hypothesis, we implemented it on a toy network, as its smaller scale allows for easier amplification and verification of this claim and on a ResNet-20. Table 7 shows the accuracy results of networks trained without the batch normalization layer.

From Table 7, it is evident that the absence of a batch normalization layer can be managed effectively by the optimization offered by INSGD. In particular, SGD diverges during training of the toy network without batch normalization, whereas INSGD achieves performance levels close to optimal settings.

Normalization layers, such as Batch Normalization, are used during forward propagation and are trainable components of neural networks. While they help stabilize the learning process and improve convergence by normalizing the input to each layer, they do not directly influence the update terms of the convolutional

Table 7: Accuracy results of the custom-designed CNN on the CIFAR-10 dataset with no batch normalization layer.

| Model | Optimizer | Initial Learning Rate | Test Accuracy |
|---|---|---|---|
| **Custom CNN** | SGD | 0.1 | 11.07±2.74% |
| | INSGD-$\ell_1$ | 0.1 | 64.85±10.98% |
| | INSGD-$\ell_2$ | 0.1 | 74.78±7.84% |
| **ResNet-20** | SGD | 0.1 | 90.14±0.18% |
| | INSGD-$\ell_1$ | 0.1 | 90.55±0.26% |
| | INSGD-$\ell_2$ | 0.1 | 90.70±0.15% |

layers. Consequently, the use of INSGD allows for normalization during backpropagation as well, further stabilizing the optimization process and potentially enhancing model performance and training efficiency.

Beyond CNNs, we trained a Vision Transformer (ViT) model on CIFAR-10 using different optimizers and settings. We hypothesized that the partitioning of an image into patches could influence how the optimizer converges during training. Table 8 presents the results of the ViT under various training configurations. To facilitate a warm startup, we employed Cosine Annealing for learning rate scheduling and trained the model for 300 epochs. The base setting utilized a batch size of 256. Additionally, we evaluated the performance of optimizers without the layer normalization layer.

Table 8: Accuracy results of the ViT on the CIFAR-10 dataset with different initial learning rates and reduction rates.

| Optimizer | LayerNorm | Learning Rate | Test Accuracy |
|---|---|---|---|
| SGD | | 0.05 | 82.79±0.40% |
| Adam | ✓ | 0.0005 | 85.38±0.44% |
| INSGD-$\ell_1$ | | 0.05 | 84.52±0.43% |
| INSGD-$\ell_2$ | | 0.05 | 84.49±0.53% |
| SGD | | 0.01 | 79.90±0.21% |
| Adam | ✗ | 0.0005 | 85.13±0.45% |
| INSGD-$\ell_1$ | | 0.05 | 83.50±0.18% |
| INSGD-$\ell_2$ | | 0.05 | 83.32±0.48% |

We observed that the INSGD optimizer can achieve convergence and results close to optimal when training the Vision Transformer (ViT). However, when trained with a warm startup, Adam outperforms both INSGD and SGD. This suggests that while INSGD can effectively handle the training of ViT models and achieve competitive results, Adam may offer superior performance under certain conditions, particularly when the learning rate is scheduled with a warm restart. Further investigation into the interplay between optimizer choice, warm startup strategies, and model architecture could provide deeper insights into the optimal training procedures for ViT models.

## 4.2 CIFAR-100 Experiment

We further extend our research by conducting experiments on the CIFAR-100 dataset. CIFAR-100 is a more challenging dataset compared to CIFAR-10 as it contains 100 classes instead of 10, requiring models to have a higher level of discrimination and classification capability. The increased class diversity in CIFAR-100 poses additional difficulty in achieving high accuracy and generalization performance. To ensure adequate representation of each class in the training process, we opted to increase the batch size to 256 for this particular experiment. Before our study, Wide ResNet-18 was recognized for its convergence capabilities and satisfactory results Zagoruyko & Komodakis (2016). In alignment with the settings outlined in the Wide

ResNet paper, we replaced the optimizer algorithm with INSGD. Similar to our CIFAR-10 experiment, the model was trained for 200 epochs, and we report the highest accuracy achieved on the testing data.

Table 9: Accuracy results of the Wide ResNet-18 on the CIFAR-100 dataset.

| Optimizer | LR | Batch | Top-1 Acc. | Top-5 Acc. |
|---|---|---|---|---|
| SGD | 0.1 | 128 | 78.24±0.43% | 94.31±0.09% |
| INSGD-$\ell_1$ | 0.1 | 128 | 78.08±0.24% | 94.34±0.17% |
| INSGD-$\ell_2$ | 0.1 | 128 | 78.47±0.16% | 94.39±0.20% |
| SGD | 0.1 | 256 | 77.22±0.43% | 93.79±0.43% |
| INSGD-$\ell_1$ | 0.1 | 256 | 78.15±0.43% | 94.54±0.43% |
| INSGD-$\ell_2$ | 0.1 | 256 | 77.89±0.43% | 93.98±0.43% |

The results presented in Table 9 provide compelling evidence of the effectiveness of the INSGD algorithm in achieving improved convergence on complex datasets. The superior performance of INSGD, as evidenced by its higher Top-1 and Top-5 accuracy, establishes its utility in training sophisticated models on challenging datasets. These findings underscore the algorithm's capability to handle intricate data distributions and optimize model performance, thereby showcasing its potential for advancing the state-of-the-art in deep learning.

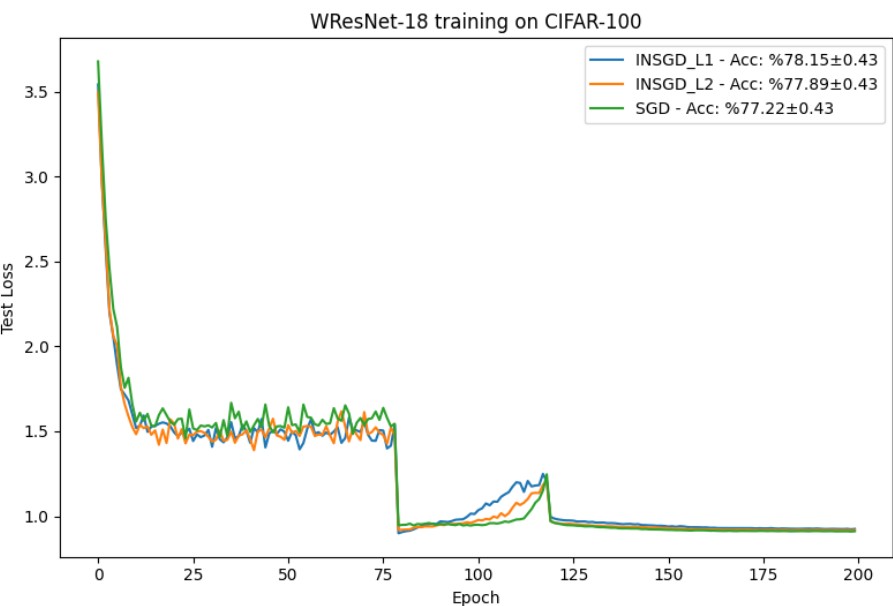

Figure 4: **Optimizer performance over 200 epochs**: Comparing test loss convergence across different optimizers.

### 4.3 ImageNet-1K Results

In this section, we present the test accuracy results on the ImageNet-1K dataset. We utilize the ResNet-50 model, as discussed in Section 3.2. The training process is conducted using the official PyTorch ImageNet-1K training code Ima (2022). Specifically, we employ the SGD and INSGD optimizers with a weight decay of 0.0001 and a momentum of 0.9.

The ImageNet-1K dataset consists of 1.2 million images and is known for its difficulty in training. Due to the image resolution and resource constraints, adopting larger batch sizes is not feasible in our environment. As a result, we train the models with a mini-batch size of 256, an initial learning rate of 0.1 for 90 epochs, and a learning rate reduction of 1/10 after every 30 epochs. Due to resource availability, we repeated the experiment 3 times with different seeds.

To augment the data, we perform random cropping and horizontal flipping with a probability of 0.5, resulting in $224 \times 224$ images. The images are then normalized using a mean of [0.485, 0.456, 0.406] and a standard deviation of [0.229, 0.224, 0.225].

The accuracy of the best models is presented in Table 10, based on the center-crop top-1 accuracy and top-5 accuracy on the ImageNet-1K validation dataset. These accuracies are obtained from the model with the highest center-crop top-1 accuracy, providing a comprehensive evaluation of the model's performance on the ImageNet-1K dataset.

Table 10: Accuracy results of ResNet-50 on the ImageNet-1K dataset.

| Optimizer | Learning Rate | Top-1 Acc. | Top-5 Acc. |
|-----------|---------------|------------|------------|
| Adam | 0.001 | 66.71±0.08% | 87.65±0.15% |
| SGD | 0.1 | 75.60±0.14% | 92.67±0.12% |
| INSGD-$\ell_1$ | 0.1 | 75.92±0.43% | 92.75±0.10% |
| INSGD-$\ell_2$ | 0.1 | 75.90±0.14% | 92.81±0.04% |

The results presented in Table 10 highlight the improved top-1 accuracy achieved by the INSGD algorithm on the ImageNet-1K dataset. This improvement is particularly significant considering the scale of the dataset, demonstrating the effectiveness of INSGD in handling large and complex datasets. By leveraging the input normalization factor, INSGD enables the model to converge more effectively by aligning the gradient direction and appropriate magnitude. Using different seeds not only confirms but also strengthens the evidence of improvement brought by the INSGD algorithm. It is essential to know that Adam may perform better training ResNet-50 however, it looks like similar settings with the INSGD and SGD, it doesn't achieve the same performance.

The power estimation obtained through momentum in INSGD indicates that the optimization algorithm can benefit from considering the entire input sequence. It suggests that the algorithm can capture long-term dependencies and utilize them for better optimization performance. Furthermore, it is worth noting that the batch size used in our experiments is relatively small compared to the number of images in the dataset. Exploring the algorithm's behavior with larger batch sizes would be an interesting avenue for future investigation.

## 5 Conclusion

In this paper, we proposed a novel neural network training method called INSGD, which incorporates ideas from the widely used NLMS algorithm in adaptive filtering. INSGD introduces a normalization step to the weight update term that normalizes the update term using only the input vector to the neurons. The normalization can be performed using both the $l_1$ and $l_2$ norms.

To evaluate the effectiveness of INSGD, we conducted experiments on various datasets using different models. Notably, our algorithm consistently demonstrated improvements in testing accuracy across multiple datasets. For example, on the CIFAR-10 dataset, INSGD achieved a significant boost in accuracy compared to traditional stochastic gradient algorithms. We observed similar positive outcomes on other datasets, such as CIFAR-100 and ImageNet-1K, when employing different models like ResNet-20 and ResNet-50.

Traditional optimization algorithms often lack flexibility when it comes to selecting hyperparameters, which can limit their effectiveness. However, the INSGD (Input Normalized Stochastic Gradient Descent) algorithm

offers solutions to this limitation by leveraging input normalization. By normalizing the input data, INSGD enables greater flexibility in tuning hyperparameters, leading to more robust and stable performance.

The promising results obtained across diverse datasets and models validate the effectiveness of INSGD in enhancing the training process. By incorporating the normalization factor into the stochastic gradient algorithm, INSGD effectively leverages the benefits of the NLMS algorithm, leading to improved performance in various object recognition scenarios. The NLMS algorithm is frequently applied in real-time processing scenarios, where the input signal exhibits temporal variations. An area worthy of exploration involves investigating the online learning mechanisms of the INSGD algorithm in such dynamic environments.

### Acknowledgments

A. E. Cetin was partially supported by NSF 2303700 and 2229659.

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
