# OpenReview forum: "Input Normalized Stochastic Gradient Descent Training for Deep Neural Networks"
_TMLR — Accepted by TMLR_

### Review · Reviewer_KDbw · 2024-04-18

**Summary Of Contributions:**

The paper introduces a novel optimization algorithm called Input Normalized Stochastic Gradient Descent (INSGD), inspired by the NLMS algorithm in adaptive filtering. The authors argue that INSGD enhances training performance for machine learning models by applying $L_1$ and  $L_2$  based normalizations to the learning rate. Experimental results show that the INSGD algorithm outperforms existing methods, improving accuracy levels on various benchmark datasets and models.

**Audience:**

Yes

**Claims And Evidence:**

Yes

**Requested Changes:**

see Strengths And Weaknesses

**Strengths And Weaknesses:**

Strengths: The authors introduce a novel algorithm for training deep neural networks.

Weaknesses: I feel that the paper's overall writing quality is subpar and lacks clarity. Several weaknesses can be identified:

Presentation and Clarity: The presentation of the paper is found to be lacking. In Section 1.2, the explanation is notably poor. The problem is redundantly defined in equations (3), (4), and (5) without subsequent utilization in the text. The introduction of the NLMS could be more succinct since a substantial amount of space is dedicated to explaining a single scheme. Furthermore, the authors introduce the update equation (24) with excessive detail, yet it is not incorporated into Algorithm 1, resulting in an overall confusing presentation.

Typographical Errors: The text contains various typographical errors. For instance, in equation (2), the minus sign ' - ' should be corrected to a plus sign ' + '. In equation (5), the variable 'w' should be represented as ${\bf w}$. Within Algorithm 1, the symbol $\tau$ should be replaced with $\gamma$. Additionally, in equation (28), $e_k$ should be altered to $w_k$.

Contribution and Theory: From my perspective, the main contribution of the paper appears to be equation (28). The authors do not adequately present the theoretical underpinnings of the paper. Therefore, it is suggested that they focus more on the numerical results. However, I believe the numeric results are inadequate as  Transformer is not included for comparison.

---

> ### Author Response · Authors · 2024-06-23
> **Response to Review**
>
> Dear Reviewer,
>
> Thank you for your comments.
>
> We have improved the writing and added new sections to enhance the quality and presentation. We believe that a clear introduction to and the inspiration behind this novel algorithm are essential for readers to understand how we derived the formula. Therefore, the detailed explanations in the first and second sections are necessary. We have also fixed the typical errors.
>
> Thank you for pointing this out. Since the algorithm is inspired by NLMS, we believe a simple theoretical analysis would be unnecessary. Instead, we demonstrate the validity of our algorithm through numerical experiments. Additionally, we have included ViT and BERT training results in the paper to expand the range of tasks.
>
> Thank you.

---

### Review · Reviewer_a8QD · 2024-04-25

**Summary Of Contributions:**

The paper introduces a novel optimization algorithm called Input Normalized Stochastic Gradient Descent (INSGD), which is inspired by the Normalized Least Mean Squares (NLMS) algorithm used in adaptive filtering. The proposed method show improvements over SGD and Adam in some medium-sized experiments.

**Audience:**

Yes

**Claims And Evidence:**

Yes

**Requested Changes:**

1.How does INSGD help "Robustness to variability"? The authors mentioned this as a motivation but did not show how it is resolved. Please verify this claim.

2. Why do need extra normalization in INSGD when we have BatchNorm in the architecture? Need discussion.

2. How would INSGD perform on non-CNN tasks, like ViT? Or on non-CV tasks such as language tasks?

**Strengths And Weaknesses:**

The paper is well written and the logic is mostly clear. The idea is interesting. Most claims in the paper are verified. I think there would be audience interested in the proposed method. One drawback is that the experiments are constrained to medium-sized CV tasks.

---

> ### Author Response · Authors · 2024-06-23
> **Response to Review**
>
> Dear Reviewer,
>
> Thank you for your review and comments. We plan to start with medium-sized computer vision tasks and hope to extend our work to different fields and tasks.
>
> Changes:
> 1. We believe that a critical issue in training any model is the selection of the hyperparameter set. Both SGD and Adam require fine-tuning of hyperparameters. One of our motivations for developing INSGD is to create an optimizer that is more robust to hyperparameter variations. We demonstrated this robustness in our experiments by varying the learning rate, batch size, and by modifying the model to exclude normalization layers.
> 2. Thank you for pointing this out. While a model utilizes a normalization layer during forward propagation and it is trainable, it does have an indirect impact on the training of other layers since INSGD uses the input for normalization. Although its impact is not as critical as in other optimizers, it can still enhance the performance of INSGD. We also added a paragraph to the paper regarding this issue.
> 3. We have added ViT and BERT training results to the paper and included performance comparisons.
>
> Thank you.

---

### Review · Reviewer_xCFE · 2024-05-27

**Summary Of Contributions:**

This paper introduces an adaptive gradient descent algorithm in which the learning rate linearly depends on the $\ell_1$ or $\ell_2$ normalization of inputs to the neuron weights for training deep learning models. Numerical results demonstrate that the corresponding $\ell_1$-SGD/$\ell_2$-SGD, or an advanced version that memorizes the past input norm (INSGD), slightly outperforms vanilla SGD across various neural network architectures.

**Audience:**

Yes

**Claims And Evidence:**

Yes

**Requested Changes:**

(1) There appears to be a typo in Equation (2) and Figure 2; the "+" sign in the equations should be a "-".

(2) The captions for tables should be placed above the tables.

(3) To be honest, I am skeptical that INSGD performs universally better than vanilla SGD. It is crucial to characterize the conditions under which INSGD can perform well, rather than solely reporting numerical results.

(4) Techniques such as Newton's method utilize approximations of the Hessian matrix to adaptively select the learning rate. I am interested in learning the authors' discussion of INSGD with these methods.

**Strengths And Weaknesses:**

Strengths: The paper is supported by extensive numerical experiments and provides a comprehensive review of related works.

Weakness: While the numerical experiments demonstrate some promise for the proposed method, the core concept of adaptively changing the learning rate has already been extensively explored in existing literature. I observed no significant advantages of the proposed algorithm and a noticeable lack of theoretical analysis, even in a basic two-layer neural network. Additionally, the paper does not provide clear guidelines on when to apply $\ell_1$-SGD versus $\ell_2$-SGD, or how to select $\beta$ in INSGD.

---

> ### Author Response · Authors · 2024-06-23
> **Response to Review**
>
> Dear Reviewer,
>
> Thank you for your review. We appreciate your comments.
>
> We believe that, since the work is inspired by the NLMS (Normalized Least Mean Squares) algorithm, theoretical analysis of the algorithm on a basic neural network would restate the validity of NLMS. Theoretical analysis for a complex CNN is not feasible mathematically, but we aim to demonstrate the impact of our algorithm through convergence graphs, using loss functions and accuracy rates. Both L1 and L2 norms can be applied in various circumstances, but we empirically observed that the L1 norm performs better when there are more outliers in the dataset. Additionally, the parameter $\beta$ can be selected based on the size of the dataset as it enables the power momentum.
>
> Changes:
> 1. The requested change is made.
> 2. The requested change is made.
> 3. We aim to show the conditions under which INSGD performs better than SGD by using various settings, such as different batch normalization, batch sizes, and learning rates. Our experiments indicate that our algorithm performs better than SGD in almost all these scenarios. However, while we believe that INSGD is more robust, we cannot claim it is universally better than SGD.
> 4. Thank you for your comment. INSGD is indeed inspired by Newton's Method and NLMS. The details can be found in (Sayed, 2005).
>
> References:
> Ali H. Sayed. Normalized LMS Algorithm, chapter 11, pp. 178–182. John Wiley & Sons, Ltd, 2008. ISBN 9780470374122. doi: https://doi.org/10.1002/9780470374122.ch18. URL: https://onlinelibrary.wiley.com/doi/abs/10.1002/9780470374122.ch18.

---

### Decision · Action_Editor_eZcK · 2024-07-02

**Recommendation:** Accept with minor revision

**Comment:**

There were many typos and the presentation suffered quite a lot in the initial submission. I understand the authors had to spend a lot of time on the extra experiments and couldn't have done a major writing revision during the discussion period. After a quick look, it appears there are still many small issues in the paper: the equation in the right side of Figure 2 uses $w_{\mathbf{y}_0}(i)$ ($w$ instead of $\mathbf{w}$); Figure 3, Figure 4, and Table 11 have missing punctuation; equation (28) wrote $log(P_k)$ instead of $\log(P_k)$; strange phrases like "by Singh et al. in Singh et al. (2015)" (why not just write "by Singh et al. (2015)"?); Table 11's top row is not bold whereas it is in the other tables; the definition of $e_k$ on page 6 is not bold, whereas everywhere else it's written as $\mathbf{e}_k$; $v(k)$ for some reason is not bold.
I'm also a bit confused as to why the authors reported only a single random seed in Figure 4. It'd be much better to provide the average over multiple seeds with confidence intervals. I also think it'd be nice to add similar figures for at least two other experiments that are not on CIFAR-10.

I hope the authors will take their time to proofread the paper, make the necessary changes, and submit the final version.

**Audience:**

Optimization is of interest to a wide audience, especially the methods that are tested on deep learning problems. All reviewers expressed the opinion that the paper has a sufficient audience.

**Claims And Evidence:**

All reviewers, except for Reviewer KDbw, selected that the paper's claims are supported by evidence. It appears that the main issue Reviewer KDbw had with the paper is with the presentation of the results and lack of empirical evaluations. However, since then, the authors have added experiments on vision transformer in Table 8. Therefore, it appears to me that the paper is not well supported by empirical evidence.

---

> ### Author Response · Authors · 2024-08-20
> **Response to Reviews**
>
> Dear Editor,
> Thank you for your recommendations and comments.
> We would like to state that we proofread the paper and made the necessary changes suggested by you. In addition, we prepared a presentation, video presentation, a link to a code repo associated with our work.
> Kind Regards